# Dynamic coupling of residues within proteins as a mechanistic foundation of many enigmatic pathogenic missense variants

**Nicholas J. Ose**[1], **Brandon M. Butler**[1☯], **Avishek Kumar**[1☯], **I. Can Kazan**[1],
**Maxwell Sanderford**[2,3], **Sudhir Kumar**[2,3,4]\*, **S. Banu Ozkan**[1]\*

**1** Department of Physics and Center for Biological Physics, Arizona State University, Tempe, Arizona, United States of America, **2** Institute for Genomics and Evolutionary Medicine, Temple University, Philadelphia, Pennsylvania, United States of America, **3** Department of Biology, Temple University, Philadelphia, Pennsylvania, United States of America, **4** Center for Genomic Medicine Research, King Abdulaziz University, Jeddah, Saudi Arabia

☯ These authors contributed equally to this work.
\* s.kumar@temple.edu (SK); Banu.Ozkan@asu.edu (SBO)

## Abstract

Many pathogenic missense mutations are found in protein positions that are neither well-conserved nor fall in any known functional domains. Consequently, we lack any mechanistic underpinning of dysfunction caused by such mutations. We explored the disruption of allosteric dynamic coupling between these positions and the known functional sites as a possible mechanism for pathogenesis. In this study, we present an analysis of 591 pathogenic missense variants in 144 human enzymes that suggests that allosteric dynamic coupling of mutated positions with known active sites is a plausible biophysical mechanism and evidence of their functional importance. We illustrate this mechanism in a case study of *β-Glucocerebrosidase* (GCase) in which a vast majority of 94 sites harboring Gaucher disease-associated missense variants are located some distance away from the active site. An analysis of the conformational dynamics of GCase suggests that mutations on these distal sites cause changes in the flexibility of active site residues despite their distance, indicating a dynamic communication network throughout the protein. The disruption of the long-distance dynamic coupling caused by missense mutations may provide a plausible general mechanistic explanation for biological dysfunction and disease.

## Author summary

Genetic diseases often occur when mutations in proteins cause gain/loss of functions. Although several methods based on conservation and protein biochemistry exist to predict genetic mutations that may impact function, many disease-associated mutations remain unexplained by these metrics. In this study, we sought a mechanistic explanation for such disease-associated mutations. In order to function, important regions of a protein must be able to exhibit collective motion. Through computer simulations, we observed that mutation of even a single amino acid position within a protein can change the protein

**Data Availability Statement:** The code to perform DFI and DCI analysis is available at https://github.com/SBOZKAN/DFI-DCI. Molecular Dynamics data are available at https://github.com/SBOZKAN/GCase. The enzyme structure list, mutation sites, and catalytic sites are contained in the Supporting information files as "S1 Table.csv". GCase disease mutation sites are contained in the Supporting information files as "S2 Table.csv". Neural net input features are contained in the Supporting information files as "S1 Data.csv". Molecular Dynamics input data are contained in the Supporting information files as "S1 Files.rar".

**Funding:** S.B.O. and N.J.O. were supported by the National Science Foundation Division of Molecular and Cellular Biosciences (award 1715591) (https://www.nsf.gov/) and the Gordon and Betty Moore Foundation Award #8415 (https://www.moore.org/) This research was supported by grants from the U.S. National Science Foundation to S.K. (GCR-1934848) (https://www.nsf.gov/) and the U.S. National Institutes of Health to S.K. (GM-139504-01) (https://www.nih.gov/) The funders had no role in study design, data collection and analysis, decision to publish, or preparation of the manuscript.

**Competing interests:** The authors have declared that no competing interests exist.

motion. We found that disease-associated mutations tend to alter the motion of regions critical to protein function, even though these mutations occur far from these critical regions. In addition, we examined the degree to which two amino acid positions within a protein may be "coupled," i.e., the extent to which motion in one position affects the other. We found that positions highly coupled to the active site of a protein are more likely to result in disease when mutated, thereby offering a new tool for predicting pathogenesis of new mutations by incorporating internal protein dynamics.

## Introduction

Our understanding of factors responsible for the pathogenesis of disease-association variants (DAVs) in proteins continues to evolve. From a biophysics perspective, it has been shown that DAVs could alter the stability of a protein [1–3]. But, only one-third of over 2,000 mutations led to a decrease in protein stability, a high-throughput functional assay revealed [4]. Rather than affecting stability, a large fraction of DAVs seems to impair specific protein-ligand function or enzymatic activity [5–8]. Furthermore, studies combining evolutionary approaches with the biochemistry of protein design have revealed that DAVs at non-conserved sites can involve complex and frequently poorly understood mechanisms [5,9–11].

Through sequencing efforts, a large catalog of missense variants in thousands of human proteins has been assembled, including those implicated in diseases (Fig 1A) [5,11–13]. However, many DAVs occur at positions that are neither evolutionary well-conserved nor a part of any known functional domain (Fig 1C). Regardless of biochemical similarity, amino acid substitutions at non-conserved sites lead to a wide range of outcomes, increasing or decreasing functional activity at up to three orders of magnitude (i.e., the rheostatic pattern of change) [14]. These enigmatic mutations are frequently misdiagnosed because neither evolutionary nor static structural features are informative. In fact, many rare missense variants occur at fast-evolving positions that do not have functional annotations (Fig 1B), which adversely impacts the prediction accuracy of commonly used methods because they run counter to expectations. In Fig 1D, we see that EvoD is able to exceed the prediction accuracy of other contemporary sequence-based metrics by accounting for additional evolutionary properties [15].

Here we explore the mechanistic role of dynamic allosteric coupling of sites carrying DAVs with the catalytic sites important for enzymatic activity. Our exploration is based on the premise that many mutations alter conformational dynamics of proteins, shifting the distribution of the ensemble and protein function, including the emergence of new functions [10,17–21], adaption to different environments [22], and dysfunction [12,23].

We use the *dynamic coupling index* (DCI) to identify sites strongly coupled to active sites critical for function [24,25]. We refer to them as dynamic allosteric residue coupling (DARC) sites. A mutation at a DARC site is likely to influence conformational dynamics and allosteric regulation, making individuals carrying mutants of these sites highly susceptible to disease phenotypes.

Firstly, in order to elucidate this allosteric mechanism, we used Molecular Dynamics (MD) simulations to examine GCase, a signature human enzyme consisting of 497 amino acids and at least 94 amino acids with observed DAVs implicated in Gaucher disease (GD) [26], which is characterized by a dangerous buildup of lipids in certain organs. Genetic changes in GCase can lead to other health conditions as well, including Parkinson's disease [27–30] and Dementia with Lewy bodies [30,31]. We investigated the mechanistic impact of these mutations on conformational dynamics and allosteric regulation [32]. In the following, we report that GD

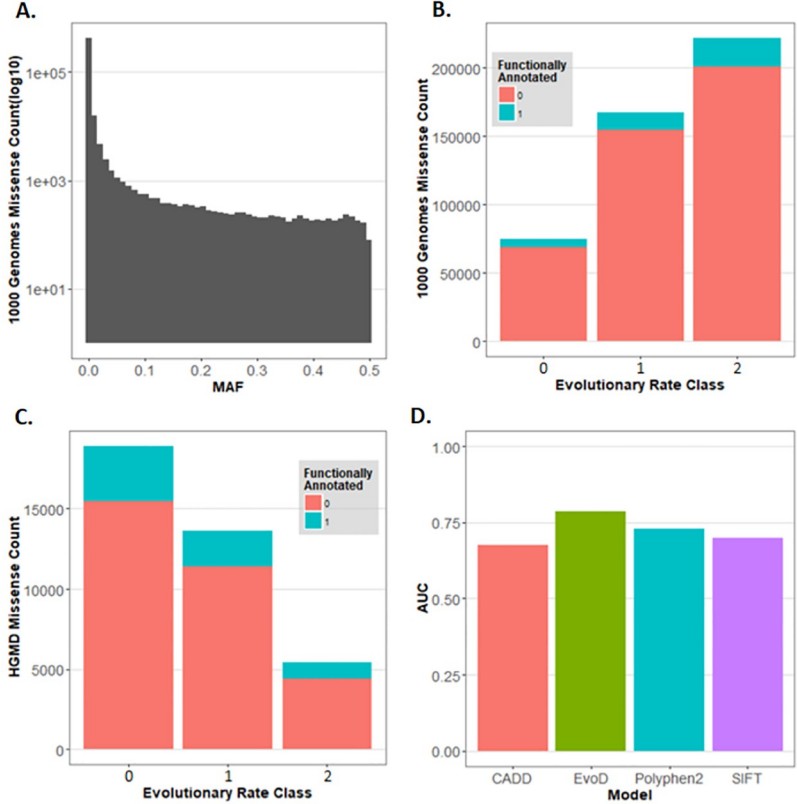

**Fig 1. Frequency, evolutionary conservation, and rates of misdiagnosis of missense variants.** (**a**) Histogram of minor allele frequencies (MAF) of missense variants in the 1000 Genomes data set. (**b**) Counts of these missense variants according to evolutionary conservation and the Uniprot functional annotation of their domain of residence. Evolutionary rate classes are from Kumar et al. [15] with class 0 sites containing no substitutions, class 1 sites exhibiting 0–1 substitutions per billion years, and class 2 sites exhibiting greater than 1 substitution per billion years. (**c**) Histogram of evolutionary conservation of sites containing only known pathogenic missense variants found in the Human Gene Mutation Database (HGMD) [16], with and without functional annotation in the Uniprot database. (**d**) Performance of four different missense diagnosis tools, quantified by their area under the receiver operation curve (AUC), which measures their ability to discriminate between putatively neutral (1000 Genomes missense variants with MAF>1%) and disease associated variants (DAVs) found in fast evolving positions (evolutionary rate class of 2). DAVs with MAF > 0.01% were excluded from these analyses.

mutations disrupt allosteric regulation due to changes in dynamic flexibility around the catalytic sites, altering enzymatic activity essential for homeostasis. The positions harboring DAVs can be thought of as key DARC sites.

While all atom MD simulations are sensitive enough to investigate how mutations at specific distal sites (usually diagnosed as benign by conventional *in silico* tools) can modulate the overall dynamics of functionally critical sites, therefore allosterically impacting function, MD approaches are often time consuming and computationally expensive. To explore the role of conformational dynamics and allostery in missense variants of many proteins with different 3-D structures at a broader scale, we utilized a more efficient coarse-grain approach, the Elastic Network Model (ENM), to conduct a proteome-wide analysis of allosteric coupling for a set of enzymes. These analyses suggest that pathogenic variants are most abundant at DARC sites. We also present an analysis of DCI asymmetry, which measures the degree of symmetry in the dynamic coupling between two sites, revealing that mutations are likely to result in a loss of function if they occur at distal sites controlled by the active site, resulting in pathogenesis.

## Results

### Disease-associated mutations modify dynamics throughout the protein

GCase is a member of the family of glycoside hydrolases that use glutamates for hydrolyzing glucocerebrosidase into glucose and ceramide. Many amino acid variants of this enzyme are reported to cause Parkinson's disease [27–30], Dementia with lewy bodies [30,31], and GD [33]. Using the crystal structure of GCase (PDB ID: 1ogs) [34] and 94 sites with DAVs [35], we calculated the Euclidean distance between the mutation site and the active site (e.g., residues 235 and 340). A vast majority (87.5%) of GD pathogenic variants occur further than 10 Å from the nearest active site residue, making direct interactions implausible. This suggests the existence of a network of indirect interactions through which a mutation at a distal site can induce dynamic changes at other regions of the protein and, by extension, impact protein function. The behavior of residues within this network can be examined by using a structural dynamics-oriented approach.

We illustrate the approach using structural dynamics by considering the example of a single mutant *N370S* that is present with a high frequency (~70%) in the Ashkenazi Jewish population and studied extensively [27,32]. We first calculated the structural flexibility profiles of residues using a position-specific dynamic flexibility index, or DFI (see the Methods section). A comparison between DFI profiles of the wild-type GCase protein with the one that contains *N370S* is shown in Fig 2A. The DFI profile provides an estimate for the role of each residue in

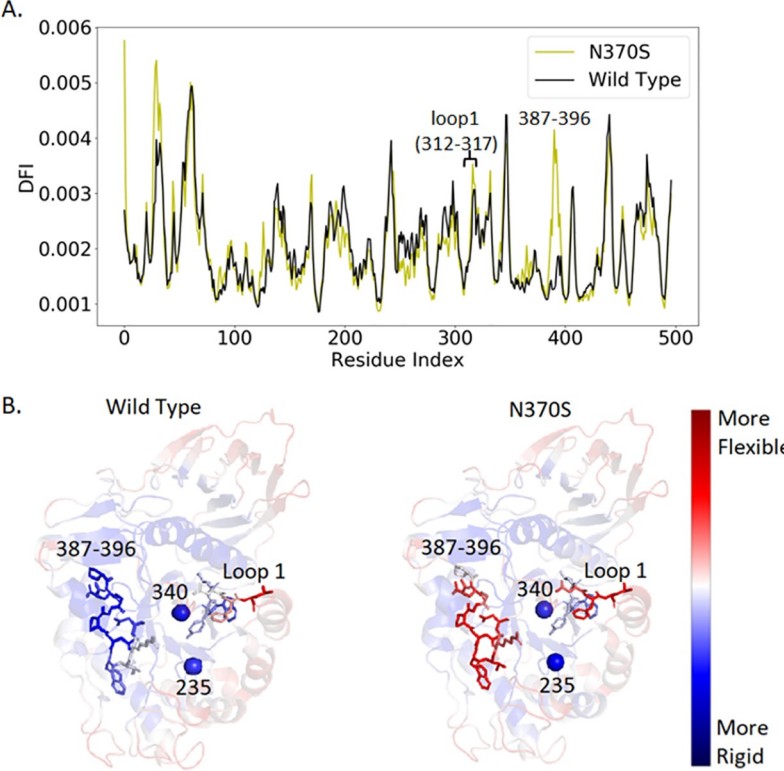

**Fig 2. A comparison of DFI profiles of wild-type GCase and N370S mutant protein.** (**a**) The %DFI profile of the mutant protein (*N370S*, yellow) is contrasted with that of the wild-type (black). Dissimilarities in the two profiles demonstrate how a single point mutation (*N370S*) can induce changes in the flexibility profile of a different region of the protein. (**b**) Ribbon diagrams showing DFI as a color-coded spectrum from red-white-blue; red and blue indicate the highest and lowest flexibility, respectively. The regions with the most significant changes in dynamic flexibility are highlighted.

mediating structure-encoded dynamics. As for GCase, the DFI profile indicates significant shifts in dynamics caused by *N370S*. Regions of the protein that should be rigid are now flexible and vice versa (Fig 2B). Hinges in the protein have moved or disappeared, and new hinges have appeared elsewhere. As reported in previous studies, these hinge shifts suggest a major change in dynamics and thus protein function [19,24,25,36].

Among the five loops surrounding the active site, we observe that loop 1 (residues 312–317) exhibits an increase in DFI scores (Fig 2A), suggesting that increased flexibility of this loop could contribute to the decrease in enzymatic activity by hindering the accessibility of the ligand to the active site as reported previously [37]. This variant displays a small change in flexibility near loop 1. Changes in DFI within loop 1 for other studied mutations are shown in the supplementary figure (S1 Fig). Additionally, the protein with *N370S* shows a very large shift in flexibility between residues 387 to 396, which overlaps with loop 3 (residues 394–399); within the overlap is the *R395* residue, which orients differently in the active and inactive states of the enzyme [23] (Fig 2A).

## Mutations at distal sites dynamically-coupled to the active site alter long-range communication

Although the only sequence difference between the wild-type and mutant GCase is a single residue, DFI changes across the protein. This behavior suggests that changes in long-range dynamic coupling may be responsible for the altered flexibility profiles. The dynamic coupling index (DCI) captures the strength of the displacement response for site $i$ upon perturbation of site $j$, relative to the average fluctuation response of site $i$ to all other sites in the protein. In this way, DCI can reveal the degree of dynamic coupling between $i$ and $j$.

Here, we present DCI as a percentile rank of the DCI range observed with values ranging from 0 to 1 (%DCI). Importantly, DFI and DCI are distinct in that DFI measures the flexibility of a position. In contrast, DCI measures the pairwise coupling of one position with another. Furthermore, DCI estimates are conditional on the functional position selected for analysis. Every amino acid position in any given protein has a unique network of direct, local interactions that give rise to a unique network of highly coupled pair positions. Across the protein structure, this gives rise to an inhomogeneous 3D interaction network. Using DCI to explore this network can be insightful when considering active sites, because it is known that even far away positions may disrupt function through the mechanism of allostery [36]. Residues that are distant enough from the active site to likely have no direct interaction (>10 Å) yet are highly coupled to them (%DCI > 60 implying a greater than average response fluctuation when active site residues are perturbed) can play an important role in protein function.

In the example of GCase, around half (52.6%) of the studied pathogenic variants, including *N370S*, occur at DARC sites (Fig 3A). In fact, according to our list of disease mutation sites [38], approximately 28% of DARC sites are associated with GD, compared to ~15% of non-DARC sites throughout the entire protein. Also, the %DCI values of DAV sites are significantly different ($P < .001$) from those of non-disease sites, as seen in Fig 3D. This suggests that variants at DARC sites are more likely to lead to genetic disease. Moreover, a comparison of DCI values of DAV sites with all other protein sites supports the same observation: mutations at DARC sites, distal sites that exhibit high coupling (i.e., high DCI), are predisposed to impact function [24,25]. Such sites may be observed in a variety of different regions and structures across a protein, as seen in Fig 3B.

Using MD simulations, we obtained the dynamic features of 20 DAVs, 2 neutral variants, and the wild-type protein. These variants were chosen because experimental data on their function was also available, by the study of Liou et al. [35]. When comparing DCI profiles of

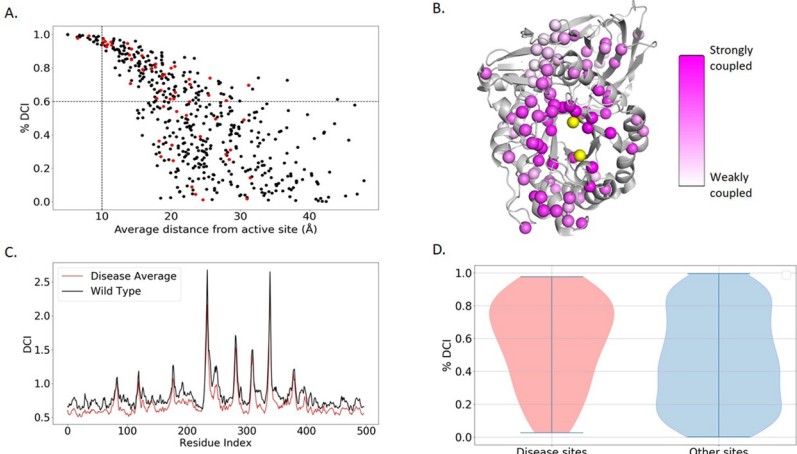

**Fig 3. DCI of GCase sites.** (**a**) Scatter plot of all GCase residues with dividers at %DCI = 60 and distance = 10 Å. The upper right quadrant contains DARC sites which can affect the active site without direct interaction. Red dots indicate severe DAVs, which have a significantly higher DCI ($P < .001$) than other sites. (**b**) A ribbon diagram showing known mutation sites of GCase (represented as pink-colored dots) and the degree of coupling to the active site delineated by the color gradient, where darker and lighter shades correspond to strongly coupled and weakly coupled, respectively. (**c**) Average DCI profile of 20 different DAVs compared to the wild-type. In general, we observe a global loss of coupling to the active site. (**d**) Violin plots showing that DAVs are generally located at sites that have higher DCI with the active site.

the active site for the wild-type and proteins with DAVs, fluctuations in DCI occur at certain sites, while mostly decreasing in GCase sites with DAVs (Fig 3C). These changes in DCI imply that the long-distance communication pathways cannot follow typical channels to the active site. This communication breakdown is presumed to be a consequence of altered dynamics. Losing rigidity in a functionally critical hinge region impairs the dynamic allosteric residue coupling, leading to a dysfunctional protein [36]. Our data also suggests a link between DCI and the severity of disease mutations. The median %DCI for DAV sites for Gaucher disease marked as "severe" was 69.6%. In comparison, mutations marked as "mild" had a median of 56.6% ($P < .045$). This further supports the idea that positions exhibiting higher dynamic coupling to the active site have a greater impact on protein function.

## Principal component analysis of DFI aligns with experimentally determined catalytic activity

As explained above, DFI profiles provide information about the dynamic function of residues throughout the protein. At the same time, DARC sites are coupled with the active site despite having no direct contact. We clustered the DFI values of DARC sites for each simulated GCase variant (Fig 4) using principal component analysis (see Methods). We found that the wild type and neutral variants (functional enzymes based on in-vitro assays) are grouped, and many of the tested proteins creating "dead enzymes" (i.e., total loss of function) are grouped as well. Liou et al. [35] used the specific activity of cross-reacting immunological material (CRIM_SA) values to estimate the catalytic rate constants ($k_{cat}$), thereby giving experiment-based estimates on the functionality of these variants. The fact that variants with higher CRIM_SA values are clustered together, as are variants with low CRIM_SA values, suggests a direct correlation between DFI profiles and CRIM_SA and, therefore, a direct correlation between DFI and protein function.

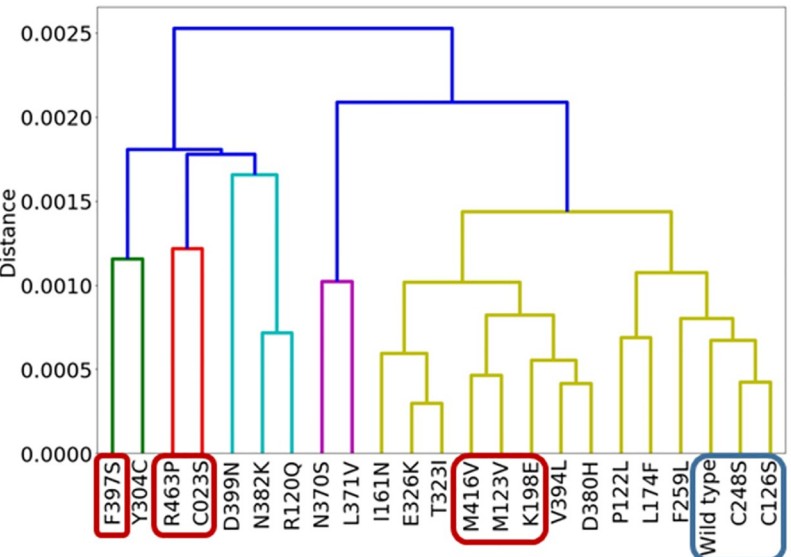

**Fig 4. Dendrograph showing clusters of GCase variants based on the DFI of DARC sites.** The variants for which experimental data is available show dead enzymes and fully functional (i.e. neutral) enzymes clustered within their own groups. Variants with CRIM_SA values of 0.3 to 1.0 are shown in blue, while variants with CRIM_SA values of 0.06 to 0.1 are shown in red. For other variants shown here, CRIM_SA values are between .1 and .3. These variants have reduced function compared to the wild type, but are still somewhat functional. Higher CRIM_SA values suggest superior enzyme function.

## A proteome-wide analysis reveals disease-associated mutations are abundant at DARC sites

After investigating GCase, we used ENM models to expand our study to include 144 human enzymes containing a total of 1024 amino acid variants (433 neutral and 591 DAVs). The ENM is a coarse grained approach and allowed us to study the dynamics of different folds efficiently. This dataset was also used in our previous work [39] incorporating the HumVar data set [40] and sequences with both a high query coverage (>80%) and sequence identity (>80%) selecting only the proteins available in the protein data bank [41]. Additionally, these protein structures had already been modeled, including any missing residues, using the Modeller software package [42].

As illustrated in Fig 5A, the DCI distribution of DAV sites shows a trend opposite to that of sites with neutral variants, exhibiting a significantly different distribution with $P < .001$. Generally, DAVs are more likely to occur at sites highly-coupled to the active site. In contrast, neutral mutations are more likely to occur at sites that are less coupled. Of the variants in this ensemble, 82% occur farther than 10 Å from the active site, suggesting that allosteric communication through 3-D network of interactions modulate the dynamics of the active site, thus impacting the function.

DCI specifically quantifies the coupling between individual positions and, as such, DCI values depend explicitly upon the positions selected for analysis. However, these pairwise interactions are not always symmetric. An interaction network may be formed such that residue perturbations may be felt more strongly in one direction than the other. If we find the difference in the DCI values between two residue positions that are not directly interacting (i.e., in spatial contact), we get a better understanding of the dynamic allostery relationship between two residues. This difference, called DCI asymmetry, provides directionality to long-distance

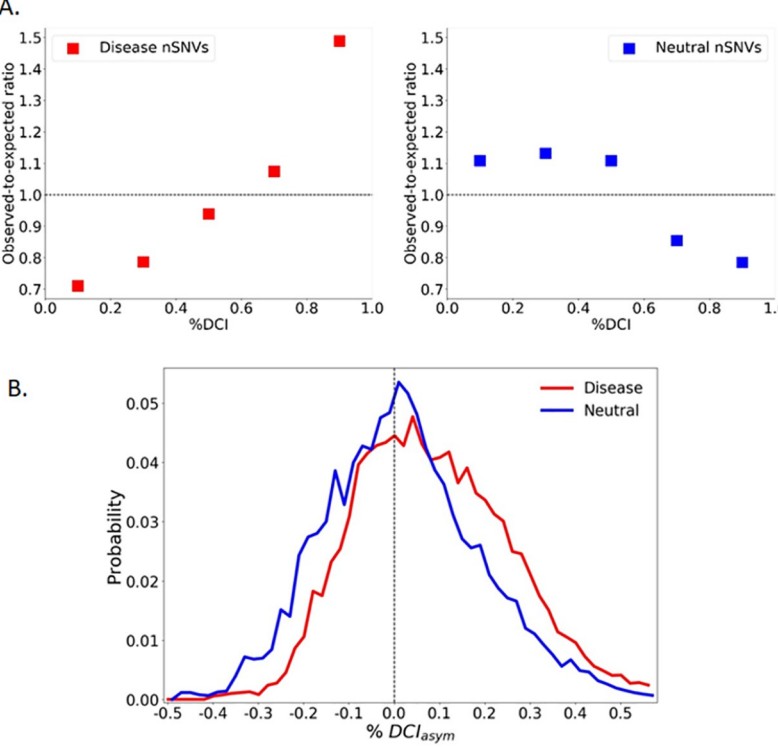

**Fig 5. %DCI and asymmetry for 144 protein ensemble. (a)** Throughout 144 proteins and 1024 variants' sites within those proteins, %DCI values were determined. These distributions were compared with the expected null distribution that %DCI values would be equally distributed over all investigates sites. Observed-to-expected ratios reveal that there are more DAVs than expected having high %DCI, whereas fewer neutral variants than expected are observed in high % DCI categories. Above the ratio equal to 1, the DAV or neutral variants occurs more often than the null expectation. Below the ratio of 1, the mutation does not occur as often as expected. **(b)** Comparison of %DCI$_{asym}$ of sites associated with neutral variants and DAVs. The distributions show a contrast as DAV sites tend to exhibit more positive values ($P$ < .001), suggesting that the active site dominates the coupling. Neutral sites on the other hand tend to give more negative asymmetry values, suggesting that the mutation site dominates. A moving average was used to visually smooth the distribution.

coupling, thereby suggesting a causal relationship In any given protein, every amino acid position has a unique network of direct, local interactions that give rise to a unique network of highly coupled partner positions [9,25,43] and heterogeneity in a 3-D network of interactions. Thus, for a particular pair of coupled amino acids (*i* and *j*), their unique network constraints differentiate the coupling of *i* to *j* from the coupling of *j* to *i*. Thus, we used the wild-type structures of our enzymes to calculate i) %DCI$_{ij}$, how strongly the position of each mutation is coupled to each active site position, ii) %DCI$_{ji}$, how strongly each active site position is coupled to the position of each mutation. From these, we calculated iii) "%DCI$_{asym}$" from (%DCI$_{ij}$–%DCI$_{ji}$) to assess the asymmetry in coupling.

Among our protein ensemble, we see a slight pattern emerge, where the interaction between disease mutation sites and active sites is generally more dominated by the active site. In contrast, the interaction between neutral mutation sites and active sites is usually dominated by the mutation site. (Fig 5B). This is indeed in agreement with our earlier findings of LacI variants [9], in which substitutions at sites where functional sites dominate the communication most often end up with a function loss.

### A neural network trained on dynamic characteristics offers superior performance at highly evolved sites

Many different methods exist to predict the effect of missense variants on protein function. Some contemporary methods focus on evolutionary considerations alongside structural information to improve the accuracy of predictions [44–47]. As one example, PolyPhen-2 uses solvent accessibility, secondary structure propensities, and crystallographic B-factors to classify mutational sites [44]. Many other approaches consider change in polarity, volume, and charge due to mutant amino acid. A number of phenotypic prediction studies use solvent accessibility, which has proven to be a useful attribute in disease prediction [46]. Other methods utilize residue–residue interaction networks of protein structures to identify functionally important residues through network topology parameters [47,48]. Evolution-based methods generally offer better performance than methods that only use structural features, yet evolution-based methods have true positive rates less than 50% for known DAVs at less-conserved positions [5,15]. In addition, their rate of correct diagnosis of true negative (benign) mutations at highly conserved positions is less than 50% [11].

Like DCI, the DFI of mutation sites can indicate an effect on protein function [36,49]. Previously, our group has shown that DFI can predict pathogenicity of protein interface sites more accurately than the accessible surface area, a commonly used metric [6]. In this study, we extended this comparison to a variety of different metrics, a larger number of missense variants, and by adding DCI in the predictive model. Using DFI, DCI, and asymmetry from our protein ensemble, we trained a neural network to predict whether certain missense variants would be neutral or not (see Methods). When used to predict the pathogenicity of random subsets of our data (90% training, 10% testing; 10-fold cross-validation), this neural network reaches the upper end of performance for established predictive software in the metrics of accuracy, precision, recall, and area under the curve (AUC) evaluated for the receiver operating characteristic (ROC) curve (Fig 6). Of particular interest is the performance of our neural network at highly evolving sites (see Methods). The evolution-based metrics tend to overestimate the rate of neutral mutations at highly-evolving sites [11,15], leading to significantly lower recall scores. However, our dynamics based approach outperforms all the other methods. This is because our method accounts for enigmatic sites—allosteric DARC sites which seem to appear neutral from an evolutionary perspective. We don't expect as many neutral mutations at those sites because we don't use any sequence information related to conservation.

Improvement is also shown over another non-evolutionary metric, Rhapsody [50,51], which is a dynamics based approach. Rhapsody utilizes a Gaussian Network Model (GNM) (a 1-D version of ENM). Thus, the major difference between the GNM based approach and our approach is that we simulate perturbation forces in three dimensions, whereas Rhapsody uses one-dimensional pairwise interactions.

## Discussion

Allostery was proposed as an important biophysical mechanism for protein function, which has led some to proclaim that allostery constitutes "the second secret of life," with the genetic code constituting "the first secret of life" [52].

Laboratory-directed evolutionary studies also highlight the emergence of mutations far from the active site [25,53,54]. These distal sites play a critical role in functional evolution, particularly in the emergence of novel functions. Yet, these distal mutation sites challenge enzyme design, as it is difficult to predict them in advance [25,55–57]. Likewise, resurrected ancestral protein studies also reveal that mutations distal from the active site are necessary for functional

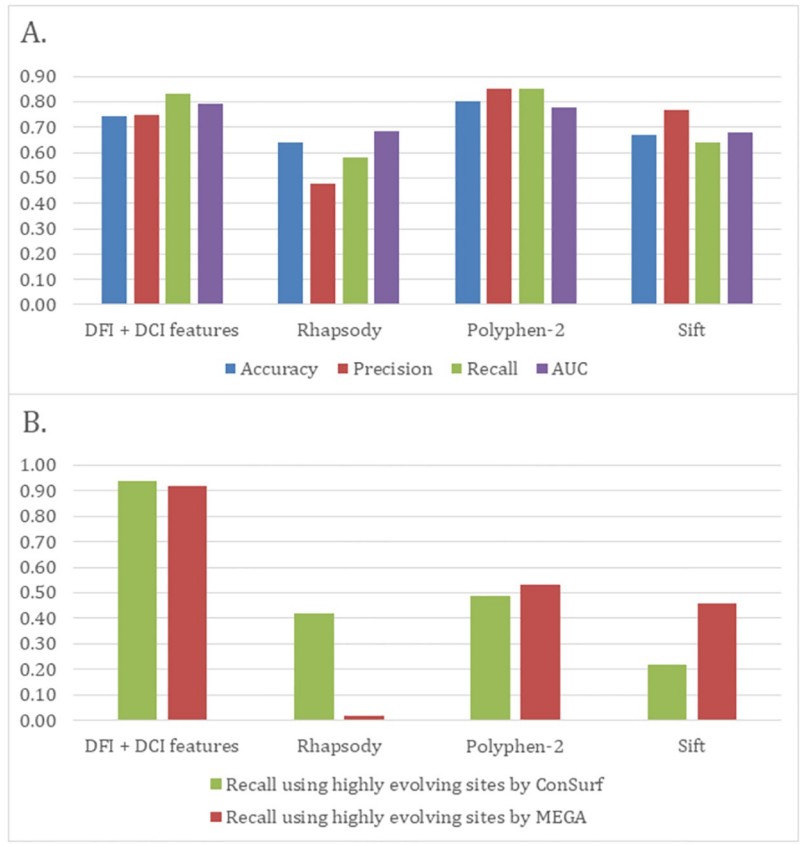

**Fig 6. Accuracy of prediction tools tested against 144 enzyme ensemble data.** (A) Bar plot showing accuracy, precision, recall, and area-under-the-curve (auc) values for four different methods including our DFI + DCI features. Without using any evolutionary data, our performance matches and may even exceed evolution-based metrics. (B) Recall values for those same metrics tested on fast evolving sites in our data. Higher false negative rates lead to lower recall values for sequence based metrics, but not for DFI + DCI features.

evolution. An example is the emergence of red color from a green ancestor in a close relative of Green Fluorescence Protein (GFP). This protein needs a minimum of 12 mutations and one deletion to convert from green to red color with high efficiency. A majority of the mutations are far from the chromophore. While the flexibility of the mutational sites does not change, in allosteric response to these mutations, both rigidification and increased flexibility occur for regions of the fold widely separated in the 3-D structure of the proteins, accommodating required flexibility for red photoconversion. These synergistic effects allow catalysis to proceed as desired and function without mutations of catalytic residue positions while maintaining fold stability and quaternary structure [19].

Here we also observe that disease-associated (i.e., function altering) mutations follow the same pattern. As neutral variants and DAVs provide the best opportunity to explore the molecular principles of how genetic variations shape phenotypic changes, we observed the same principle of dynamic allostery such that functions become altered through distal mutations while conserving the amino-acid sequence of catalytic residues. We have found that the disruption of the allosteric dynamics with functionally-important sites in a protein is a mechanistic explanation for many missense variants associated with diseases and other biological phenotypes. The patterns of dynamic coupling with the active sites are different for disease and neutral phenotypes for missense mutations that occur at spatially-distant positions to

functional (active) sites. Specific analysis of GCase proteins also provides evidence of the same mechanism observed in resurrected studies. These distal mutations allosterically modify flexibility profiles of different sites, leading to a change in function.

This finding also suggests that rather than affecting only protein stability, the disruption of ligand binding, or both, the allosteric dynamic coupling and stability explain how a large fraction of disease-associated variants impair protein-ligand function or enzymatic activity [6,7,12]. A high-throughput functional assay of over 2,000 variants also show that only a minority of mutations led to a decrease in protein stability [4]. Thus, our findings align with the neutral theory of molecular evolution, as mutations on functionally important catalytic sites must have been eliminated by negative selection due to critical functional loss. On the other hand, the distal mutations remotely fine-tune the native state ensemble to modify function without interfering with folding/folding stability.

We are in the era of rapid development of next-generation methods for whole-genome, whole-exome, and targeted sequencing that has produced an unprecedented amount of data. Among all the genetic variation data, the most commonly observed variants are missense, and identifying the missense variants with pathogenic effects that contribute to disease or drug sensitivities is the primary goal of 21st-century genomic analysis and phylomedicine. As stated in a review of allostery by Liu and Nussinov [52], uniting the genetic code, which constitutes "the first secret of life," and allostery, "the second secret of life," could reveal a generalized disease mechanism and allow for the discovery of novel drugs, as well as blueprints for innovative personalized treatment methods.

## Methods

### Dataset

A total of 144 individual monomeric protein structures from the Protein Data Bank (PDB) [41] were collected from a BLAST search of sequences with requirements of ≥80% sequence identity and ≥80% query coverage to ensure only structures that could be accurately mapped to human variation data were included. Human genetic variations were obtained from the HumVar, and HumDiv databases [38] with 1024 amino acid variants, where 433 were neutral and 591 were deleterious.

### Determining catalytic sites

The catalytic sites were gathered from the Catalytic Site Atlas (CSA) database [58], which identifies the residues directly involved in catalyzing the reactions of enzymes. Since these residues are critical for protein function, they were used as input into our dynamic coupling index (DCI) metric. The entries in the CSA were either "original entries" derived from the literature itself or "homology entries" based on sequence comparison with the literature-based original entries. In either case, the catalytic sites purported by the CSA should accurately represent functional sites on the protein. Our dataset contained 144 enzymatic proteins that mapped to entries in the CSA database.

### Calculating functional-dynamics profiles

Dynamic flexibility index (DFI) quantifies the dynamic stability of a given position. It measures the resilience of a position to perturbations initiated at positions in the protein distal to the residue in question, but to which it is linked via structurally encoded global dynamics. Therefore, DFI profiles provide important information about protein function. Namely, residues that exhibit very low DFI scores (DFIs) do not show large amplitude fluctuations in

response to random Brownian kicks but rather transfer the perturbation energies throughout the chain in a cascade fashion; examples of low *DFI* residues are those in hinge regions. Hinges are parts of the protein which are generally rigid. At the same time, they do not exhibit a high fluctuation response to perturbations but transfer these perturbations to the rest of the protein. Like hinges on a door, they stand still, providing an anchor point for other parts to move around.

The method for obtaining the dynamic flexibility index (DFI) is based on the perturbation response scanning (PRS) method [59], in which the C-alpha atom of each residue in the protein is modeled as a node in an elastic network model (ENM). The interaction between each node is modeled by a harmonic potential with a distance-dependent spring constant [59,60]. A small perturbation in the form of an external random force (i.e., Brownian kick) is sequentially applied on each node in the network, and the perturbation response of all nodes is recorded according to linear response theory as

$$[\mathbf{\Delta R}]_{3N \times 1} = [\mathbf{H}]^{-1}_{3N \times 3N}[\mathbf{F}]_{3N \times 1} \tag{1}$$

where $\mathbf{F}$ is the external random force, $\mathbf{H}^{-1}$ is the inverse Hessian, and $\mathbf{\Delta R}$ is the positional displacement of all $N$ nodes in three dimensions.

However, ENM is a coarse-grained model. To improve the accuracy of this model and allow sensitivity to mutations, the hessian inverse can be replaced with the covariance matrices obtained from molecular dynamics simulations.

$$[\mathbf{\Delta R}]_{3N \times 1} = [\mathbf{G}]_{3N \times 3N}[\mathbf{F}]_{3N \times 1} \tag{2}$$

Here, G is the covariance matrix containing the dynamic properties of the system. The covariance matrix contains the data for long-range interactions, solvation effects, and biochemical specificities of all types of interactions.

Each perturbation is performed in ten different directions to ensure an isotropic response. The perturbation is repeated for every node in the network, and the positional displacements $\mathbf{\Delta R}$ of each node are stored in a perturbation matrix $\mathbf{A}$ given by

$$[\mathbf{A}]_{N \times N} = \begin{bmatrix} \Delta|R^1|_1 & \Delta|R^2|_1 & \cdots & \Delta|R^N|_1 \\ \Delta|R^1|_2 & \Delta|R^2|_2 & \cdots & \Delta|R^N|_2 \\ \vdots & \vdots & \ddots & \vdots \\ \Delta|R^1|_{N-1} & \Delta|R^2|_{N-1} & \cdots & \Delta|R^N|_{N-1} \\ \Delta|R^1|_N & \Delta|R^2|_N & \cdots & \Delta|R^N|_N \end{bmatrix} \tag{3}$$

where $|\Delta R^j|_i = \sqrt{\langle (\Delta R)^2 \rangle}$ is the magnitude of the positional displacement of each residue $i$ in response to a perturbation at residue $j$. The DFI score of residue $i$ is defined as the sum of the total displacement of residue $i$ induced by a perturbation on all residues, which is computed by taking the sum of the $i$-th row of the perturbation matrix $\mathbf{A}$,

$$\mathrm{DFI}_i = \frac{\sum_{j=1}^{N} |\Delta R^j|_i}{\sum_{i=1}^{N} \sum_{j=1}^{N} |\Delta R^j|_i} \tag{4}$$

where the denominator is the total displacement of all residues, used as a normalizing factor. Therefore, the greater the DFI score at position $i$, the more flexible that site will be and the lower the score, the more rigid that site will be, meaning it has less of a response to perturbations throughout the protein. Oftentimes it can be useful to examine the flexibility of certain

residues relative to the flexibility range of that single protein. To do this, DFI values can be ranked on a percentage scale as shown below:

$$\%DFI_i = \frac{n_{\leq i}}{N} \tag{5}$$

where $n_{\leq i}$ is the number of positions having $DFI \leq DFI_i$.

Recently, we have extended this method to identify allosteric links or dynamic coupling between any given residue and functionally important residues by introducing a new metric, the *dynamic coupling index* (DCI) [36]. The DCI metric can identify DARC sites, which are distal to functional sites but control them through dynamic allosteric coupling. This type of allosteric coupling is important; sites with strong dynamic allosteric coupling to functionally critical residues (DARC sites), regardless of separation distance, likely contribute to the function. Thus, a mutation at such a site can disrupt the allosteric dynamic coupling or regulation, leading to functional degradation. As defined, DCI is the ratio of the sum of the mean square fluctuation response of the residue $i$ upon functional site j perturbations (i.e., catalytic residues) to the response of residue $i$ upon perturbations on all residues. DCI enables us to identify DARC site residues, which are more sensitive to perturbations exerted on residues critical for function. This index can be utilized to determine residues involved in allosteric regulation. It is expressed as

$$DCI_{ij} = \frac{\sum_{j=1}^{N_{functional}} |\Delta R^j|_i / N_{functional}}{\sum_{j=1}^{N} |\Delta R^j|_i / N} \tag{6}$$

where $|\Delta R^j|_i$ is the response fluctuation profile of residue $i$ upon perturbation of residue $j$. The numerator is the average mean square fluctuation response obtained over the perturbation of the functionally critical residues $N_{functional}$. The denominator is the average mean square fluctuation response over all residues. Just as with DFI, DCI may also be ranked on a percentage scale:

$$\%DCI_{ij} = \frac{m_{\leq i}}{N} \tag{7}$$

where $m_{\leq i}$ is the number of positions having $DCI \leq DCI_{ij}$.

We further investigated the change in dynamics upon mutation compared to the wild type structure using ΔDFI and ΔDCI. The delta-DFI (ΔDFI) profile was calculated as

$$\Delta DFI_i = \frac{DFI_{disease} - DFI_{wt}}{DFI_{wt}} \tag{8}$$

Where $DFI_{disease}$ is the dynamics profile for the mutated protein structure and $DFI_{wt}$ is the dynamics profile for the wild-type structure. Similarly, the delta-DCI (ΔDCI) profile was calculated as

$$\Delta DCI_i = \frac{DCI_{disease} - DCI_{wt}}{DCI_{wt}} \tag{9}$$

One additional tool we use is DCI asymmetry, which measures preferential information transfer through asymmetric dynamic coupling. Simply put, the coupling asymmetry between

positions i and j can be calculated as

$$DCI_{asym} = DCI_{ij} - DCI_{ji} \tag{10}$$

$$\%DCI_{asym} = \%DCI_{ij} - \%DCI_{ji} \tag{11}$$

Where $DCI_{ij}$ represents the relative response of residue i to a perturbation at residue j and $DCI_{ji}$ represents the relative response of residue j to a perturbation at residue i.

It should be once again made clear that all dynamic analysis of the GCase protein was conducted using data from MD simulations only, while analysis of the 144 enzyme ensemble was performed using data from ENM simulations only.

## Molecular dynamics simulations

To compute the DFI and DCI profiles of each missense variant of GCase, we first performed MD simulations to obtain the native ensemble of each variant and then applied our analysis. The starting structure for GCase was taken from the Protein Data Bank (accession number 1ogs [34]). The mutagenesis tool was used in Pymol [61] to create variant structures. Next, we loaded structures into TLEAP using the ff14SB force field [62]. We then added protein hydrogens were and a 14.0 Å cubic box of TIP3P surrounding water atoms, followed by Na$^+$ and Cl$^-$ atoms for neutralization [63]. Then all systems were energy-minimized using the SANDER module of AMBER 14 [64,65]. First, the protein was kept fixed with harmonic restraints to allow surrounding water molecules and ions to relax, followed by a second minimization step in which the restraints were removed and the protein-solution was further minimized. Both minimization steps employ the method of steepest descent followed by conjugate gradient.

We then ran heating, density equilibration and production using the GPU-accelerated PMEMD module of AMBER 14 [65]. Periodic boundary conditions were used in all simulations, and the bond lengths of all covalent hydrogen bonds were constrained using SHAKE [64]. Direct-sum, non-bonded interactions were cut off at distances of 9.0 Å or greater, and long-range electrostatic interactions were calculated using the particle mesh Ewald method [66,67]. During the heating cycle, we heated systems from 0K to 300K over a duration of 250 ps. The density of the system was then allowed to equilibrate over 5 ns at constant temperature and pressure. A Langevin thermostat was used to control the temperature at 300 K and a Berendsen barostat to adjust the pressure at 1 bar. We used a timestep of 2 fs and saved structural conformations every 10 ps. All simulations were allowed to progress to 1 μs of total simulation time, deemed the minimal required simulation time for convergence based on earlier studies [24,68].

In order to calculate DFI and DCI, we calculated covariance matrices using 50 ns moving windows that overlap by 25 ns over the last 500 ns of the trajectory of each simulation. In order to ensure ergodicity where the DFI and DCI profiles present the equilibrium dynamics, there are two of the basic conditions that need to be met: (i) All conformations must be sampled from the same distribution. (ii) The time windows and subsequent covariance matrices obtained ought to be independent of the initial atomic coordinates in order to eliminate global motions and accurately capture equilibrium coordinate information. Because of this, the final average DFI profiles will be independent of the window size; meaning that the averaging of DFI profiles from different time window sizes (i.e. 50 ns vs 75 ns) will give similar results and the calculated covariance matrices extracted from different times of trajectories should also result in similar DFI profiles, such as seen in S2 Fig.

## Clustering the DFI values of DARC sites

We clustered the DFI profiles of DARC sites for various mutated GCase proteins by comparing their percentile rankings. To compare the flexibility profiles, the proteins are concatenated into a data matrix $X$. The statistical procedure Singular value decomposition (SVD) is used to factorize the data into the orthonormal basis, which is a representation of the vector space containing data. It is similar to principal component analysis which may be used to assist in understanding the structure of data or to increase the signal-to-noise ratio in data by eliminating the redundant dimensions and mapping it on a lower-dimensional space. Clustering by SVD acts as an effective noise filter by isolating the highest variances among data points in the top principal vectors. Consequently, the remaining insignificant singular vectors can be omitted from the reconstruction.

The DFI profiles of all proteins are merged into a matrix $X$, of dimensions ($m \times n$). Here $m$ is the number of datasets (protein variants) we are clustering together, each having $n$ number of attributes (n = number of DARC sites, thus each element in a given column presents the DFI value of specific DARC site of a given variant). On performing SVD, $X$ is decomposed as follows:

$$[X]_{m \times n} = [U]_{m \times m}[\zeta]_{m \times n}[V]_{n \times n} \tag{12}$$

Here, $U$ and $V$ are unitary matrices with orthonormal columns and are called left singular vectors and right singular vectors, respectively, and $\zeta$ is a diagonal matrix with diagonal elements known as the singular values of $X$.

The singular values of $X$, by convention, are arranged in a decreasing order of their magnitude; $\sigma = \{\sigma_i\}$ represent the variances in the corresponding left and right singular vectors. The set of highest singular values representing the largest variance in the orthonormal singular vectors can be interpreted to show the characteristics in the data $X$ and the right singular vectors create the orthonormal basis which spans the vector space representing the data. The left singular vectors contain weights indicating the significance of each attribute in the dataset as $w_i = \sum_{k=1}^{r} \sigma_k |u_{ik}|$. Using these features of the decomposed singular vectors, we can create another matrix, $X^*$ using only the highest '$r$' singular values which can mimic the basic characteristics of the original dataset. Thus, $X^*$ can be represented as

$$[X^*]_{m \times r} = [V^*]_{m \times r}[\zeta^*]_{r \times r} \tag{13}$$

Here, $\zeta^*$ contains only largest $r$ singular values and $V^*$ contains the corresponding right singular vectors. The data are now clustered hierarchically based on the pairwise distance between different protein variants in the reconstructed DFI data with reduced dimensions.

For a pair of datasets (or between flexibility profiles of any two proteins) $j_1$ and $j_2$, the distance between them in the original set of data was given by

$$d_{12} = \sqrt{\sum_{i=1}^{n} (X_i^{j1} - X_i^{j2})^2} \tag{14}$$

which in reduced dimensions can be calculated as

$$d_{12} = \sqrt{\sum_{i=1}^{r} (X_i^{*j1} - X_i^{*j2})^2} \tag{15}$$

These pairwise distances are used as the parameters for clustering the flexibility profiles of GCase. The DFI values of DARC sites are aligned and clubbed into a dataset matrix $X$. The three largest singular values are used for reconstruction of data and clustering. The pairwise distance between each protein using the equation above is used for clustering them hierarchically.

A bottom-up approach is used for the hierarchical clustering, where initially each protein variant is assigned its own cluster and then, in successive iteration, closest clusters are merged together into a common cluster. In this approach, the distance between clusters is defined by the average pairwise distance between their components (average linkage clustering [69]). In the end, the clusters are represented hierarchically using a dendrogram, where the vertical axis denotes the Euclidean distance between various clusters and among their sub-clusters.

## Neural network

In an attempt to enhance our prediction accuracy based on protein dynamics we integrated a Neural Network based training and prediction algorithm. With an increased number of dimensions in data space, regular regression methods fall behind machine learning strategies and artificial Neural Networks. Our data contains multiple dynamics driven metrics emerging from per position specific DFI of the position with observed variant and also DFI of the neighborhood positions as well as DCI. These metrics by themselves display strong correlation (Fig 5) [49], but proteins are dynamic systems meaning per residue dynamics cannot grant all the relative information about the global dynamics. Therefore, with the inclusion of several distinct metrics that represent different dynamical features of the proteins, we exploited an Artificial Neural Net based prediction approach.

The feed forward Neural Network architecture deployed in this paper utilizes a single input layer with multiple features and a binary classification model. The features include: $DFI_i$, $\%DCI_{ji}$, $DCI_{asym}$, and the average DFI of residues within 7Å. We use residues within 7Å because they have direct interactions with the variant site. These four features along with corresponding sites and ground truth values may be found in the Supplementary Materials as S1 Data. The network includes two hidden layers with 80 nodes each between the input and the output layer. The hidden layers are connected with a 50% dropout scheme to eliminate overfitting. The initial node weights and biases of the network are sampled from a uniform distribution with Rectified Linear Units as the activation function to reach better convergence compared to a sigmoid function. The output layer has initial uniform node weights and biases sampled from Xavier uniform initializer and a sigmoid activation function with binary label output. The optimization algorithm utilizes a stochastic gradient descent with built-in momentum to minimize the cross-entropy loss function. The built-in momentum helps to escape saddle points and reach a global minimum loss. The learning rate for the optimizer is set as 0.001 with 1000 epochs in total for the Neural Network to converge. The Neural Network is trained with 90% of randomly selected data points and tested by the remaining 10%. This process is repeated 10 times to gather improved statistics and eliminate any bias coming from the data itself. Employing a 10-fold training/testing algorithm provides a distribution of accuracies instead a single accuracy.

The evaluation metrics AUC, accuracy, precision and recall are used to evaluate the predictive power of the classification model by comparing prediction values with the ground truth values. The four possible outcomes from the binary classifier are: True positive (TP), true negative (TN), false positive (FP), and false negative (FN). Accuracy, precision, and recall equations for calculation are denoted below:

$$accuracy = \frac{TP + TN}{TP + TN + FP + FN}$$

$$precision = \frac{TP}{TP + FP}$$

$$recall = \frac{TP}{TP + FN}$$

AUC is obtained by calculating the area under the Receiver Operating Characteristic (ROC) curve, which is generated using true positive rate and false positive rates.

In order to determine highly evolving sites, we utilized two different methods: (i) the ConSurf database to evaluate conservation of each site, [70,71], and (ii) Molecular Evolutionary Genetics Analysis (MEGA) software [72] to calculate evolutionary rates as described by Kumar et al. [11].

## Supporting information

**S1 Fig. Stick diagrams of loop 1 which are colored corresponding to their DFI for 19 different disease variants.** DFI here is a color code within a spectrum of red-white-blue where red shows the highest, and blue shows the lowest flexible sites.
(TIF)

**S2 Fig. %DFI profiles averaged over different time scales demonstrate convergence.** Black: average %DFI values calculated using covariance matrix data over 400ns to 600ns of the wild type GCase simulation. Blue: average %DFI values calculated using covariance matrix data over 600ns to 800ns of the wild type GCase simulation. Red: average %DFI values calculated using covariance matrix data over 800ns to 1ms of the wild type GCase simulation. All profiles use 50 ns moving windows that overlap by 25 ns when calculating average %DFI.
(TIF)

**S3 Fig. Accuracy of prediction tools tested against highly evolving sites in 144 enzyme ensemble data.** Bar plots showing accuracy, precision, recall, and area-under-the-curve (auc) values for four different methods including our DFI + DCI features. (A) The prediction methods were evaluated using only fast evolving sits according to ConSurf. (B) The prediction methods were evaluated using only fast evolving sits according to MEGA. Note that for our highly evolving subset, rhapsody returned 0 true positive and 0 false negative values, causing AUC, precision, and recall to be either zero or incalculable. Using either set of highly evolving sites, we are slightly better in AUC and comparable in precision. However, the dynamics based classifier have slightly lower values for accuracy owing to higher false positive rates.
(TIF)

**S1 Table. Enzyme ensemble information.** For each missense variant used in our analysis, shows the PDB identification code, mutation site, pathogenicity (disease or neutral), and active sites. Sites are aligned to the associated PDB file.
(CSV)

**S2 Table. Shows the mutation site of each GCase DAV used in our analysis.** Asterisk denotes sites with multiple DAVs reported. Sites are aligned to PDB ID: 1ogs [34].
(CSV)

**S1 Data. Input data for our neural network (see Methods).** *dfi_i*, *%dci_ji*, *dci asymmetry*, and *average dfi within 7Å* columns contain input layer features and the *disease(1) or neutral(0)* column contains ground truth values for pathogenicity. Columns *pdb id* and *pdb residue index* exist for identification purposes.
(CSV)

**S1 Files. Contains input files for MD simulations of GCase variants.**
(RAR)

## Author Contributions

**Conceptualization:** Nicholas J. Ose, Brandon M. Butler, Avishek Kumar, Sudhir Kumar, S. Banu Ozkan.

**Data curation:** Nicholas J. Ose, Brandon M. Butler, Avishek Kumar, I. Can Kazan, Maxwell Sanderford, Sudhir Kumar.

**Formal analysis:** Nicholas J. Ose, I. Can Kazan, Maxwell Sanderford, Sudhir Kumar, S. Banu Ozkan.

**Funding acquisition:** Sudhir Kumar, S. Banu Ozkan.

**Investigation:** Nicholas J. Ose, Brandon M. Butler, Avishek Kumar, Maxwell Sanderford, Sudhir Kumar.

**Methodology:** Nicholas J. Ose, Brandon M. Butler, Avishek Kumar, I. Can Kazan, Sudhir Kumar, S. Banu Ozkan.

**Project administration:** S. Banu Ozkan.

**Resources:** S. Banu Ozkan.

**Software:** Nicholas J. Ose, Avishek Kumar, I. Can Kazan.

**Supervision:** Sudhir Kumar, S. Banu Ozkan.

**Visualization:** Brandon M. Butler, Avishek Kumar, Sudhir Kumar, S. Banu Ozkan.

**Writing – original draft:** Nicholas J. Ose, Brandon M. Butler, Avishek Kumar, Sudhir Kumar, S. Banu Ozkan.

**Writing – review & editing:** Nicholas J. Ose, Brandon M. Butler, Avishek Kumar, I. Can Kazan, Sudhir Kumar, S. Banu Ozkan.

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
