## [Decision Letter · Decision Letter 0]

4 Nov 2021

Dear Professor Ozkan,

Thank you very much for submitting your manuscript "Dynamic coupling of residues within proteins as a mechanistic foundation of many enigmatic pathogenic missense variants" for consideration at PLOS Computational Biology.

As with all papers reviewed by the journal, your manuscript was reviewed by members of the editorial board and by several independent reviewers. In light of the reviews (below this email), we would like to invite the resubmission of a significantly-revised version that takes into account the reviewers' comments.

In this case it is extremely important that you show that your conclusions applies generally and not only to one (or a handful) of examples, as pointed out by the reviewers

We cannot make any decision about publication until we have seen the revised manuscript and your response to the reviewers' comments. Your revised manuscript is also likely to be sent to reviewers for further evaluation.

Sincerely,

Anders Wallqvist

Associate Editor

PLOS Computational Biology

Arne Elofsson

Deputy Editor

PLOS Computational Biology

Reviewer's Responses to Questions

**Comments to the Authors:**

Reviewer #1: In the manuscript submitted by Nicholas J. Ose, et al, with title “Dynamic coupling of residues within proteins as a mechanistic foundation of many enigmatic pathogenic missense variants”, the authors presented a computational study utilizing dynamic coupling index (DCI) and dynamic flexibility index (DFI) to analyze disease related mutations at dynamic allosteric residue coupling (DARC) sites of 144 human enzymes containing 591 pathogenic missense variants. The dynamical correlations among these mutations distal from the enzymatic active sites are systematically evaluated and compared. The manuscript is well organized and presented and should be published after minor revisions to address the following concerns.

1. Figure 5 and its caption are not clear or easy to follow. For example, term “nSVNs” is not defined anywhere in the manuscript.

2. In the title, it is indicated that Principal component analysis (PCA) was carried out for DFI values of DARC sites. But the PCA seems not to be presented in the section, or at least it is not clear.

Reviewer #2: Ose et al. present a computational study mapping genetic missense mutations to phenotypes. Primary tools include the dynamic coupling index or DCI, which measures the allosteric coupling between two residue sites. The method exerts random external forces on the functional site (e.g., the active site) and monitors responses of other (distal) sites. Residues with larger responses, called DARC sites, are speculated to be more important for the function. The DCI is based on the linear response theory established about 15 years ago and having gained further development in multiple recent studies. Using DCI, the authors have examined disease mutations in the human acid beta-glucosidase and 144 other human enzymes with annotated disease/non-disease mutations. They found that the predicted DARC sites have a higher potential to cause a loss-of-function effect leading to diseases. Disease mutations caused a change of flexibility near the active site and an overall reduction in the dynamic coupling, suggesting the functional relevance of the metric.

Identifying potential disease mutations has a great impact in improving human health. In this paper, the authors try to establish the mechanistic basis for the genotype-phenotype relationship. The question to be addressed is clearly important and some of the correlations are certainly quite interesting.

Incorporating dynamics in predicting disease mutations is not completely new. For example, Ponzoni and Bahar, PNAS 2018. The use of DCI indeed brings some insights, but the significance of the findings is unclear because of the lack of connections with previous studies. For example, how will the authors compare their method with the metrics used in the 2018’s and many other previous studies? Will DCI show a stronger correlation? What about the predictive power (e.g., evaluated using AUC) compared to the current state-of-the-arts? Some discussion on the potential complementary nature of DCI to other sequence-based, structural, or dynamic metrics will also be helpful.

A main argument here is the correlation between DCI and disease mutations. However, a correlation does not guarantee a deterministic role. According to Fig. 3D, there are still plenty of disease sites associated with low DCIs and, more importantly, many non-disease sites associated with high DCIs. The correlation may suggest a functional relevance but does not necessary establish that the metric is deterministic or even a primary factor.

The overall flow of the manuscript is clear, but there are some missing details that affect the clarity to some extent. For example:

The description of MD simulations (in Methods) is too short and clearly does not meet the current standard for transparency and reproducibility.

It is unclear if the authors performed MD for all the mentioned proteins (including >90 GCases and 144 human enzymes) or just for GCases and used ENMs for other enzymes.

The principal component analysis is mentioned but there is no result. Was PCA used in the clustering?

What were the 20-ns simulation windows (p. 20) used for?

Minor points:

In the abstract, it says 94 GD mutants but in the Results it becomes 200 (p. 5) and then 97 (p. 6). Please check the number and be consistent.

Fig. 1. What is the difference between b & c? Are they from different databases? Why are the trends opposite?

P. 4, line 69. ‘…, which adversely impact the prediction accuracy of commonly used methods because they run counter to expectations (Fig 1d).’ This sentence is unclear and needs to be further explained.

P. 8, line 167. ‘DCI measures the coupling of a position’. It should be ‘coupling of two positions.’

Fig. 4 legend. It should be ‘DARC sites.’

Not all figure panels are cited in the text.

Reviewer #3: The work presented particularly exciting. The possibility of having a structural explanation for the impact of a point mutation involved in pathology and particularly interesting. This work follows on from other research carried out either on a specific protein or on a large set of data. (State of the art could be deeper).

This work at the border of these two categories and this makes reading the manuscript slightly difficult. I had to reread the entire manuscript several times to be sure what I was reading comma was either the protein involved in Gaucher disease or it was a large number of proteins. I am not sure, moreover, that I have followed everything correctly.

Thus figure 1 presented, quickly in the manuscript, relates to a large data set point when Figure 2 it is only on the protein of Gaucher disease. The figures are not analysed for the specialist. It is difficult to know if the results are significant or not. As an example figure 2A is composed of 1 example, one small region seems different, but not at the point of mutations, but no statistical analysis allow to see it. How is it on the other SNPs?

And, a general question arises on the use of SNPs associated with this pathology, is it possible to have, thanks to the different projects of 100000 genomes++, all the non-pathological SNPs and then obtain -in fact- exactly the same results. It is a necessary that must be in this paper.

The work is mainly based on an existing methodology, which must nevertheless be defended in a more rigorous way. It does not seem to be very sensitive. It would be advisable to better integrate its explanation and its critical analysis in the whole of the manuscript.

The choice of this enzyme seems to be more related to the distance from the active site as a problem for its function as allosteric questioning, at long range. Figure 3A represents difficulty of reading it is difficult to see what is really relevant from what is not. The black dots and red dots are not separable. The choice of threshold values is not explain. It is difficult to make an opinion.

We would like to have other examples with different shapes and folds.

There is clearly work. It is particularly unfortunate not to be able to evaluate it correctly because of the presentation of the manuscript and especially a rather too strong absence of the critical aspect on the results.

Reviewer #4: In this manuscript, the authors compare disease mutations with neutral mutations in human enzymes, in terms of their allosteric dynamic coupling with known enzymatic active sites. The computation analysis is based on a combination of elastic network models and molecular dynamics simulation. The authors conduct both case studies and proteome-wide analyses, concluding that disease mutations tend to disrupt catalytic function through dynamic allosteric coupling with active sites.

On Pages 6-7, in the section entitled "Disease-associated mutations modify dynamics throughout the protein", the authors investigate a single disease mutation N370S in the enzyme GCase, and show that this disease mutation leads to an increase in flexibility (as measured by DFI) within or near loop 1 and/or loop 3. Are the authors suggesting that for the GCase enzyme, disease mutations on average lead to higher DFI values within or near loop 1 and/or loop 3 than neutral mutations? If so, this assertion should be rigorously tested with p-values presented. If not, the authors should clearly describe the conclusions from their analyses.

In general, the manuscript contains numerous general assertions regarding disease versus neutral mutations, only some of which are supported by p-values. The authors should support their general assertions by p-values whenever possible.

The definition and application of the DCI metric are somewhat confusing. DCI is defined to measure the impact of catalytic site perturbation on a residue under investigation. Here, catalytic site is the cause, and the residue under investigation is the consequence. However, the authors then apply the DCI metric to identify and study allosteric residues, where presumably the residue under investigation is the cause, and the catalytic site is the consequence. What is the rationale for applying DCI in this context, given that the direction of cause and effect seem to be reversed?

On Page 10, the first section is entitled "Principal component analysis of DFI aligns with experimentally determined catalytic activity". However, in this section the authors only performed clustering analysis (Fig. 4), but not principal component analysis.

On Page 19, Equation (5) contains several errors. The division sign should be "/" rather than "\\". In the numerator, the index j should sum over from 1 to N_functional, rather than from N_functional to N_functional. In the denominator, the index beneath the summation symbol should be j rather than i.

On Page 20, Equation (9), the equation "DCI_asymm = DCI_i - DCI_j" does not make sense and should be fixed and further elaborated.

It is not clear if the authors have made all data underlying their findings fully available. The authors should try to make their data as fully available as possible. The data can either be provided as supporting information, or deposited to a public repository.

Minor comments and typos:

Page 5, Line 98: "wcontaining" -> "containing".

Page 7, Fig. 2 mentions "%DFI profile", but "%DFI" is not defined.

Page 12, Line 273: "sit" -> "site".

Fig. 3a, 3b, and 3d are not referred to in the manuscript text.

**Have the authors made all data and (if applicable) computational code underlying the findings in their manuscript fully available?**

Reviewer #1: Yes

Reviewer #2: Yes

Reviewer #3: Yes

Reviewer #4: None

PLOS authors have the option to publish the peer review history of their article (what does this mean?). If published, this will include your full peer review and any attached files.

Reviewer #1: No

Reviewer #2: No

Reviewer #3: No

Reviewer #4: No
---

## [Decision Letter · Decision Letter 1]

7 Feb 2022

Dear Professor Ozkan,

Thank you very much for submitting your manuscript "Dynamic coupling of residues within proteins as a mechanistic foundation of many enigmatic pathogenic missense variants" for consideration at PLOS Computational Biology. As with all papers reviewed by the journal, your manuscript was reviewed by members of the editorial board and by several independent reviewers. The reviewers appreciated the attention to an important topic. Based on the reviews, we are likely to accept this manuscript for publication, providing that you modify the manuscript according to the review recommendations.

Sincerely,

Anders Wallqvist

Associate Editor

PLOS Computational Biology

Arne Elofsson

Deputy Editor

PLOS Computational Biology

[LINK]

Reviewer's Responses to Questions

**Comments to the Authors:**

Reviewer #1: The authors have addressed all the concerns raised by this reviewer. Now the manuscript should be accepted for publication.

Reviewer #2: The authors have done great work to revise their paper and have addressed all my previous concerns. I only have two additional minor points to mention regarding the new text, which I believe the authors can fix easily:

It would be better to have a brief explanation of the scores (accuracy, recall, etc.) used in the benchmark.

Fig. 6B, only recalls are shown. Better to show other scores as well for completeness (maybe in the supplemental file).

Reviewer #3: I am particularly and pleasantly surprised by the quality of the responses given to all the reviewers. The authors have taken all the comments into account and wanted to answer all the questions in depth. They did this with great success, which gave me a much better understanding of this work. It deserves to be published as it is and I hope will have the impact it deserves.

Reviewer #4: All comments have been adequately addressed.

**Have the authors made all data and (if applicable) computational code underlying the findings in their manuscript fully available?**

Reviewer #1: Yes

Reviewer #2: Yes

Reviewer #3: Yes

Reviewer #4: None

PLOS authors have the option to publish the peer review history of their article (what does this mean?). If published, this will include your full peer review and any attached files.

Reviewer #1: No

Reviewer #2: No

Reviewer #3: No

Reviewer #4: No

Figure Files:

Data Requirements:

Reproducibility:

References:

---

## [Editor Report · Decision Letter 2]

9 Mar 2022

Dear Professor Ozkan,

We are pleased to inform you that your manuscript 'Dynamic coupling of residues within proteins as a mechanistic foundation of many enigmatic pathogenic missense variants' has been provisionally accepted for publication in PLOS Computational Biology.

Best regards,

Anders Wallqvist

Associate Editor

PLOS Computational Biology

Arne Elofsson

Deputy Editor

PLOS Computational Biology

---

## [Editor Report · Acceptance letter]

4 Apr 2022

PCOMPBIOL-D-21-01712R2 

Dynamic coupling of residues within proteins as a mechanistic foundation of many enigmatic pathogenic missense variants

Dear Dr Ozkan,

I am pleased to inform you that your manuscript has been formally accepted for publication in PLOS Computational Biology. Your manuscript is now with our production department and you will be notified of the publication date in due course.

With kind regards,

Livia Horvath
